# Distribution Matching Variational AutoEncoder

**Sen Ye** [1]   **Jianning Pei** [2]   **Mengde Xu** [3]   **Shuyang Gu** [3]   **Chunyu Wang** [3]   **Liwei Wang** [1 4]   **Han Hu** [3]

## Abstract

Most visual generative models compress images into a latent space before applying diffusion or autoregressive modelling. Yet, existing approaches such as VAEs and foundation model aligned encoders implicitly constrain the latent space without explicitly shaping its distribution, making it unclear which types of distributions are optimal for modeling. We introduce **Distribution-Matching VAE** (**DMVAE**), which explicitly aligns the encoder's latent distribution with an arbitrary reference distribution via a distribution matching constraint. This generalizes beyond the Gaussian prior of conventional VAEs, enabling alignment with distributions derived from self-supervised features, diffusion noise, or other prior distributions. With DMVAE, we can systematically investigate which latent distributions are more conducive to modeling, and we find that SSL-derived distributions provide an excellent balance between reconstruction fidelity and modeling efficiency, reaching a gFID of 3.2 on ImageNet with only 64 training epochs. Our results suggest that choosing a suitable latent distribution structure (achieved via distribution-level alignment), rather than relying on fixed priors, is key to bridging the gap between easy-to-model latents and high-fidelity image synthesis. Code is available at https://github.com/sen-ye/dmvae.

## 1. Introduction

The field of visual synthesis has made staggering progress, with generative models now capable of producing high-resolution, photorealistic images from complex text descriptions (Ramesh et al., 2022; Saharia et al., 2022). A dominant paradigm underpinning these successes is a two-stage generation process (Rombach et al., 2022; Gu et al., 2022). This approach first employs a powerful autoencoder, or "tokenizer," to compress a high-dimensional image $x$ into a compact, low-dimensional latent representation $z = E(x)$. Subsequently, a powerful generative model, such as a diffusion model (Ho et al., 2020; Song et al., 2020) or an autoregressive transformer (Esser et al., 2021; Ramesh et al., 2021), is trained to model the prior distribution $p(z)$ of these latents.

Despite this success, the structure of the latent space remains largely a black box. A critical unresolved issue is that current methods provide no guarantee that the resulting latent distribution is suitable for the second-stage generative model. Without explicit control over the global geometry of the latent space, these approaches often lead to a latent space which is either too complex and irregular for the generative model to learn effectively, or it is too rigidly constrained, sacrificing the high-fidelity details required for reconstruction.

Existing strategies fail to resolve this dilemma because they do not operate at the distributional level. Standard VAEs (Rombach et al., 2022) (Figure 1a) enforce a KL divergence constraint, but this penalty operates on individual samples. It ignores the properties of the aggregate distribution, failing to ensure that the global latent geometry is aligned with the generative model's capability. RAE-like designs (Zheng et al., 2025) (Figure 1b) bypass this by using a *fixed* pre-trained encoder to extract structured features; while these features are easier to model, the encoder cannot be optimized for reconstruction, leading to significant loss of visual details. Similarly, pointwise matching methods (Yao et al., 2025; Chen et al., 2025a) (Figure 1c) attempt to align latent codes to target features (e.g., SSL embeddings). However, like VAEs, they constrain only individual samples and neglect the collective density of the latents, leaving the aggregate distribution structure unregulated. Fundamentally, these limitations stem from a lack of explicit control over the aggregate posterior distribution, defined as $q(z) = \int q(z|x)p(x)\,dx$. Without regulating this integral, it is impossible to precisely sculpt the latent space or to rigorously test how different prior structures causally

---

[1]State Key Laboratory of General Artificial Intelligence, School of Intelligence Science and Technology, Peking University [2]University of Chinese Academy of Sciences [3]Tencent [4]Center for Machine Learning Research, Peking University. Correspondence to: Shuyang Gu <cientgu@tencent.com>, Liwei Wang <wanglw@pku.edu.cn>.

*Proceedings of the 43rd International Conference on Machine Learning*, Seoul, South Korea. PMLR 306, 2026. Copyright 2026 by the author(s).

affect downstream generative performance.

In this paper, we propose **Distribution Matching VAE (DMVAE)**, a framework designed to bridge this gap (Figure 1d). DMVAE generalizes the VAE objective by replacing the standard KL-divergence with a flexible distribution matching constraint. By leveraging Distribution Matching Distillation (DMD) (Yin et al., 2024b), we utilize a diffusion model to estimate the score of an arbitrary reference distribution $p_r(z)$ and force the encoder's aggregate posterior $q(z)$ to align with it, which distills the structure of $p_r(z)$ into $q(z)$. This transforms the autoencoder from a black box into a controllable probe: we can now mold the latent space into any desired shape—whether it be a Gaussian, text embeddings, or SSL features—and rigorously evaluate which structure maximizes generative efficiency.

Leveraging DMVAE, we conduct a first large-scale systematic study of latent priors. We evaluate a spectrum of reference distributions, including Gaussian distributions, text-embedding distributions (SigLIP (Zhai et al., 2023)), supervised features (ResNet (He et al., 2016)), and self-supervised features (DINO (Caron et al., 2021; Oquab et al., 2023)).

Our investigation yields a decisive finding: the choice of latent distribution is the primary driver of convergence speed and modeling quality. We discover that SSL-derived distributions (specifically DINO) are optimal for diffusion modeling. SSL distributions possess a semantic clustering that is inherently easier for diffusion models to learn, while retaining sufficient richness for high-fidelity reconstruction.

This finding translates into significant efficiency gains. By aligning the latent space with the optimal SSL prior, our model drastically reduces the training budget required for high-quality generation. In summary, our contributions are:

1. We propose Distribution Matching VAE (DMVAE), a framework that aligns the latent aggregate posterior to any pre-defined reference distribution via distribution matching, enabling explicit control over latent structure.
2. We conduct the first systematic study of latent priors, evaluating how different distribution forms (e.g., Gaussian vs. SSL features) impact the learnability of the subsequent diffusion model.
3. We demonstrate that selecting the correct prior (SSL features) leads to state-of-the-art convergence efficiency. DMVAE achieves a gFID of 3.22 on ImageNet 256x256 with only 64 training epochs, and 1.82 with 400 epochs.

## 2. Preliminary

We first define the notation used throughout the paper. Let $x \in \mathcal{X}$ be an input image from the data distribution $p(x)$. A tokenizer consists of an encoder $E_\theta : \mathcal{X} \to \mathcal{Z}$ that maps the image to a latent code $z = E_\theta(x)$, and a decoder $G_\omega : \mathcal{Z} \to \mathcal{X}$ that reconstructs the image $\hat{x} = G_\omega(z)$.

### 2.1. Variational Autoencoders (VAEs)

The standard $\beta$-VAE (Kingma & Welling, 2013) framework trains the encoder $E_\theta$ (parameterizing $q_\theta(z|x)$) and decoder $G_\omega$ (parameterizing $p_\omega(x|z)$) by maximizing the Evidence Lower Bound (ELBO):

$$\mathcal{L}_{\text{VAE}} = \mathbb{E}_{p(x)} \Big[ \mathbb{E}_{q_\theta(z|x)}[\log p_\omega(x|z)] \\ - \beta \, D_{\text{KL}}(q_\theta(z|x) \| p(z)) \Big]. \tag{1}$$

where $p(z)$ is a fixed, simple prior, typically a standard Gaussian $\mathcal{N}(0, I)$. The KL-divergence term acts as a regularizer, forcing the per-sample posterior $q_\theta(z|x)$ to be close to $p(z)$, which in turn simplifies the aggregate posterior $q(z)$.

While a large $\beta$ was initially intended for strong regularization to ensure generative capacity, it frequently causes model collapse. Consequently, in latent diffusion models which use VAE as tokenizer, $\beta$ is often set to a very small value (e.g., $1 \times 10^{-6}$ in (Rombach et al., 2022)).

### 2.2. Flow Matching

This paper uses diffusion/flow-based models as distribution estimators in the latent space. We adopt the standard forward noising process: given a clean latent $z_0 \sim p(z)$, sample $t \sim \mathcal{U}[0, 1]$ and $\epsilon \sim \mathcal{N}(0, I)$, and construct

$$z_t = \alpha_t z_0 + \sigma_t \epsilon, \tag{2}$$

where $\{\alpha_t, \sigma_t\}$ are fixed noise schedules.

We parameterize the distribution via a time-conditioned velocity network $v(\cdot, t)$. The flow matching objective (Lipman et al., 2022) is defined as:

$$\mathcal{L}_{\text{FM}}(v; p) = \mathbb{E}_{t, z_0, \epsilon} \left[ \| v(z_t, t) - (\epsilon - z_0) \|_2^2 \right] \tag{3}$$

In addition, we use the corresponding *score* notation $s(z_t, t) = \nabla_{z_t} \log p_t(z_t)$. For common SDE/ODE parameterizations, the score can be obtained from the velocity by a known linear transformation (see, e.g., (Ma et al., 2024)). Throughout the paper, we keep both notations: $v(\cdot, \cdot)$ for training with $\mathcal{L}_{\text{FM}}$ and $s(\cdot, \cdot)$ for expressing distribution-matching gradients.

### 2.3. Inject distribution prior into VAEs

Recent works have explored moving beyond a simple Gaussian prior.

**VAVAE** (Yao et al., 2025; Chen et al., 2025a) explicitly aligns latent codes $z = E_\theta(x)$ with features from a pre-trained DINO model, denoted as $\phi(x)$, via an auxiliary

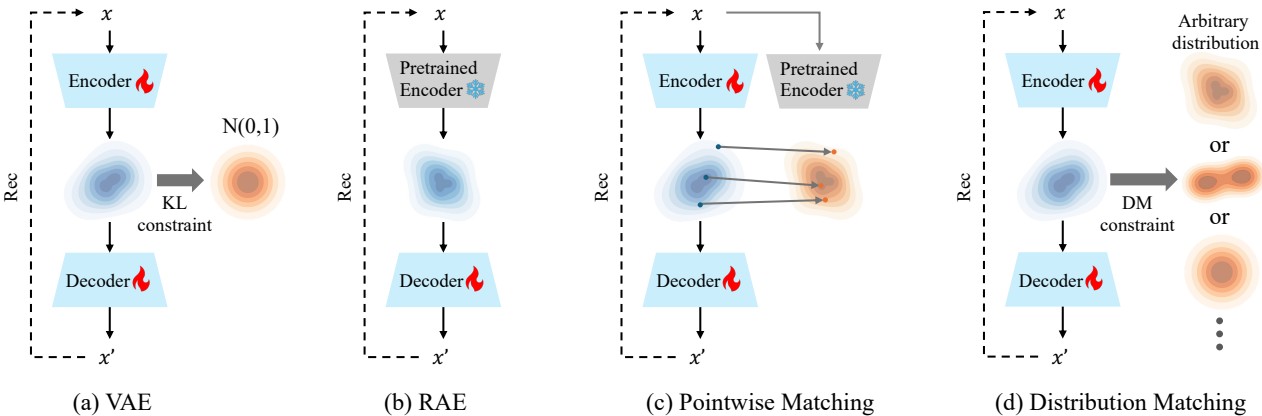

(a) VAE       (b) RAE       (c) Pointwise Matching       (d) Distribution Matching

*Figure 1.* Illustration of VAE (Kingma & Welling, 2013), RAE (Zheng et al., 2025), pointwise matching encoder (Yao et al., 2025; Chen et al., 2025a), and Distribution Matching VAE.

alignment loss:

$$\mathcal{L}_{\text{VAVAE}} = \mathcal{L}_{\text{recon}} + \lambda \left\| E_\theta(x) - \text{sg}[\phi(x)] \right\|_2^2, \quad (4)$$

where $\text{sg}[\cdot]$ denotes the stop-gradient operator. This encourages $E_\theta(x)$ to inherit semantic properties of $\phi(x)$.

**Diffusion Priors** (Li et al., 2025) attempt to use a powerful pre-trained diffusion model as a prior in latent space. A representative approach (also referred to as a "flow prior") updates the encoder by backpropagating a diffusion/flow-matching style objective through $z_0 = E_\theta(x)$, which is known to be unstable and can lead to posterior collapse (Poole et al., 2022).

**Adversarial Autoencoders (AAE)** (Makhzani et al., 2015) use a GAN-based objective to shape the latent distribution. An auxiliary discriminator $D$ distinguishes samples from the aggregate posterior $q(z)$ and a reference prior $p_r(z)$, while the encoder is trained to fool $D$:

$$\mathcal{L}_{\text{AAE}} = \mathcal{L}_{\text{recon}} +$$
$$\min_E \max_D \mathbb{E}_{z \sim p_r(z)}[\log D(z)] + \mathbb{E}_{x \sim p(x)}[\log(1 - D(E(x)))]$$
$$(5)$$

While AAEs allow $q(z)$ to match a chosen $p_r(z)$, they are limited by GAN training instability and the expressive capacity of $D$. Our work instead leverages diffusion/flow models to enable stable matching to complex reference distributions.

## 3. Distribution Matching VAE

**The Pitfall of Per-Sample Regularization.** Our goal is to gain explicit control over the aggregate posterior distribution $q(z) = \int q(z|x)p(x)dx$. Previous methods (Yao et al., 2025; Chen et al., 2025a) attempt this indirectly via per-sample regularization, which is fundamentally insufficient. The classic VAE loss (Equation (1)) regularizes the

per-sample posterior $q(z|x)$. While this forces each $q(z|x)$ towards the prior $p(z)$, it places no explicit constraint on the mixture of these posteriors, $q(z)$. If the decoder is powerful, the encoder can learn to map different inputs $x$ to distinct, well-separated modes, (i.e., $q(z|x)$ collapses to a delta function). The resulting aggregate posterior $q(z)$ becomes a complex, multi-modal, and "holey" manifold (Rezende & Viola, 2018; Razavi et al., 2019), which is notoriously difficult for a generative prior $p(z)$ to model. Similarly, methods like VAVAE (Chen et al., 2025a) (Equation (4)) use a per-sample distance loss, which only guarantees that each $E(x)$ is near its corresponding $\phi(x)$, but does not constrain the global structure of the $q(z)$ manifold.

### 3.1. The DMVAE Objective

Our goal is to gain explicit control over the encoder-induced *aggregate posterior* $q(z) = \int q(z|x)p(x) \, dx$ by matching it to an arbitrary reference distribution $p_r(z)$. Both $q(z)$ and $p_r(z)$ are implicit and high-dimensional, making divergences such as $D_{\text{KL}}(q(z)\|p_r(z))$ intractable to evaluate directly.

**Key idea: score-based distribution matching with an online student.** A distribution can be characterized by its time-dependent score under the diffusion/flow noising process. We therefore represent the reference distribution $p_r(z)$ by a frozen teacher score model $s_{\text{real}}$. However, directly pre-computing (and distilling) a fixed score for $q(z)$ is not viable, because $q(z)$ changes continuously as the encoder learns. This motivates an *online* student score model $s_{\text{fake}}$ that tracks the evolving $q(z)$ during training, following the joint-training principle of Distribution Matching Distillation (DMD) (Yin et al., 2024b).

**Step 1: pre-train the reference (teacher) score model.** We train a time-conditioned velocity network $v_{\text{real}}(\cdot, t)$ on samples from the fixed reference distribution $z_0 \sim p_r(z)$ by minimizing the flow matching loss in Eq. (3). After convergence,

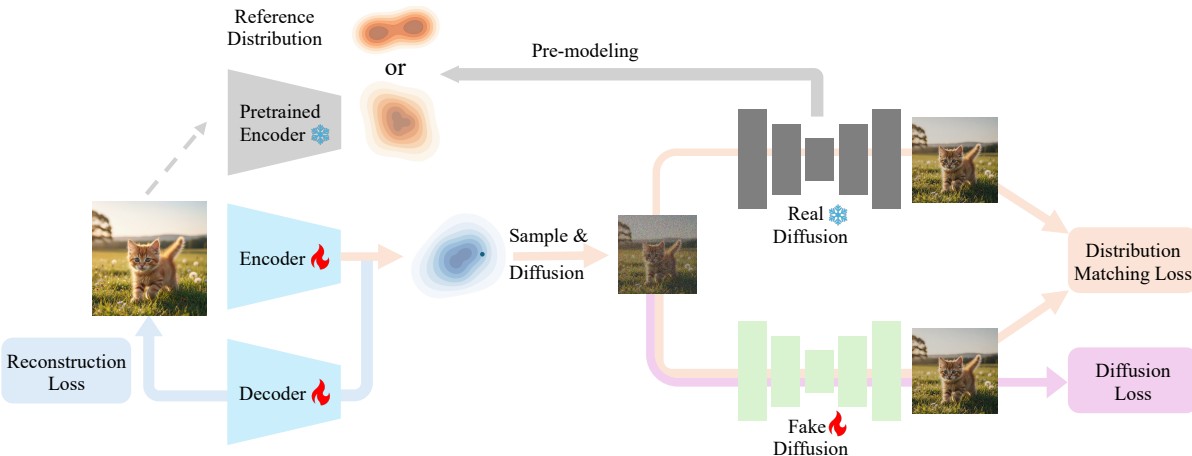

*Figure 2.* The training pipeline of Distribution Matching VAE.

$v_{\text{real}}$ is frozen and serves as the canonical estimator of $p_r(z)$. We obtain the teacher score $s_{\text{real}}(z_t, t) = \nabla_{z_t} \log p_{r,t}(z_t)$ from $v_{\text{real}}$ via a known linear transformation under the chosen SDE/ODE parameterization (Ma et al., 2024).

**Step 2: joint training of VAE and the student (fake) score model.** We jointly train the VAE $(E_\theta, G_\omega)$ and a student velocity/score model $(v_{\text{fake}}, s_{\text{fake}})$. Each iteration consists of three coupled updates with clearly separated gradient flows:

**(i) Reconstruction update (updates $\theta, \omega$).** Given $x \sim p(x)$, we encode $z_0 = E_\theta(x)$ and reconstruct $\hat{x} = G_\omega(z_0)$. We update $(\theta, \omega)$ using the reconstruction loss

$$\mathcal{L}_{\text{recon}} = \mathbb{E}_{x \sim p(x)} \left[ d\big(x, G_\omega(E_\theta(x))\big) \right]. \qquad (6)$$

**(ii) Student score update (updates only the fake model).** The student model is trained to estimate the score/velocity of the *current* aggregate posterior induced by $E_\theta$. Importantly, we stop gradients through the encoder so that this objective updates only $v_{\text{fake}}$:

$$\mathcal{L}_{\text{fm}} = \mathbb{E}_{t,\epsilon,z_0} \left[ \|v_{\text{fake}}(z_t, t) - (\epsilon - \text{sg}[z_0])\|_2^2 \right] \qquad (7)$$

**(iii) Distribution matching update (updates $\theta$ only).** To align $q(z)$ with $p_r(z)$, we update the encoder so that the student score matches the teacher score. Following (Yin et al., 2024b), we implement this via a gradient estimator that does not backpropagate through the student score network:

$$\nabla_\theta \mathcal{L}_{\text{DM}} \approx \mathbb{E}_{t,x,\epsilon} \left[ w_t \left( s_{\text{fake}}(z_t, t) - s_{\text{real}}(z_t, t) \right) \frac{dE_\theta(x)}{d\theta} \right], \qquad (8)$$

where $z_t = \alpha_t E_\theta(x) + \sigma_t \epsilon$, and $w_t$ balances different noise levels (Yin et al., 2024b). Note that $\mathcal{L}_{\text{DM}}$ is not required in closed form; Eq. (8) specifies the update direction we implement.

**Overall optimization.** The overall training pipeline is shown in Figure 2 and we summarize training as minimizing

the following objective:

$$\mathcal{L}_{\text{total}} = \mathcal{L}_{\text{recon}} + \gamma \mathcal{L}_{\text{fm}} + \lambda \mathcal{L}_{\text{DM}}, \qquad (9)$$

**Per-Sample vs. Distributional Constraints.** The distribution matching formulation (Equation (9)) is fundamentally different from VAEs (Equation (1)) or VAVAE (Equation (4)). The key difference lies in how they behave under the inevitable tension between the reconstruction loss $\mathcal{L}_{\text{recon}}$ and the prior constraint.

Per-sample losses, such as the VAE's KL-divergence or VAVAE's pairwise distance, optimize an *expectation* of a local loss: $\mathbb{E}_x[\mathcal{L}(E_\theta(x), ...)]$. This objective can be deceptively minimized. Consider a special case: For half the data $\{x_A\}$, $\mathcal{L}_{\text{recon}}$ dominates and $\mathcal{L}_{\text{align}}$ dominates the other half $\{x_B\}$. Under the joint optimization, suppose the latents $E_\theta(x_A)$ remain on the original reconstruction manifold $\mathcal{M}_{\text{recon}}$, and $E_\theta(x_B)$ are pulled perfectly onto the prior $p_r(z)$. In this case, the *total average alignment loss* $\mathbb{E}[\mathcal{L}_{\text{align}}]$ is halved. However, the aggregate posterior $q(z)$ has *degenerated* into a disjoint mixture $q(z) \approx 0.5 \cdot \mathcal{M}_{\text{recon}} + 0.5 \cdot p_r(z)$. This topologically damaged manifold is arguably *worse* for generative modeling than the original $\mathcal{M}_{\text{recon}}$, despite the seemingly low average aligned loss.

In contrast, our DMVAE objective optimizes a *global geometric divergence* $D_{\text{KL}}(q(z) \| p_r(z))$. This loss cannot be deceptively minimized in this way. The score function $\nabla \log q(z)$ of the disjoint mixture distribution described above is globally different from the (presumably smoother) score $\nabla \log p_r(z)$ of the target prior. Therefore, $s_{\text{fake}}$ (which must track this complex score) will fail to match $s_{\text{real}}$, causing the distribution matching loss to remain high. Even if $\mathcal{L}_{\text{recon}}$ prevents a perfect match, $\mathcal{L}_{\text{DM}}$ will always push $q(z)$ towards a holistic structural alignment with $p_r(z)$, resulting in a denser, non-holey manifold that is far more suitable for

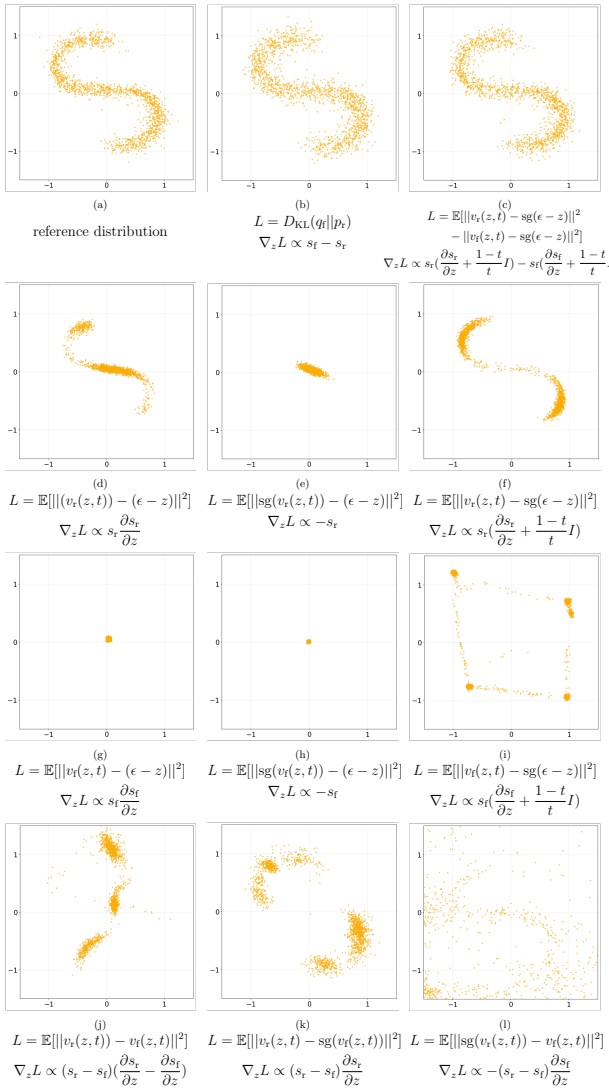

**Figure 3.** Analysis of different distribution matching objectives on a 2D setting. (a) illustrates the reference distribution; (b) denotes the distribution matching objective, (d,e,f) represent real score maximization with different stopping gradients; (g,h,i) represent methods for fake score maximization; (j,k,l) represent directly optimizing the difference between real and fake scores; Finally, (c) represents optimizing the difference between the real and fake diffusion losses. For each objective, we have listed the loss function and their gradient $\nabla_z \mathcal{L}$.

a generative prior.

## 3.2. Variations of Distribution Matching Objective

To investigate the optimal objective for aligning the aggregate posterior $q(z)$ with the reference prior $p_r(z)$, we analyze several distinct gradient fields derived from different score-based objectives. We visualize the learned latent distribution in a 2D toy experiment by these objectives, as shown in Figure 3, and mathematically analyze their gradi-

ents to understand their convergence behaviors. For brevity, we omit time-dependent weights $w_t$ and the chain rule term $\frac{dE_\theta(x)}{d\theta}$, focusing on the gradient field $\nabla_z \mathcal{L}$ that acts on the latent samples $z = E_\theta(x)$.

**Real Score Maximization.** This objective is also referred to as a "Flow Prior" (Li et al., 2025), maximizes the likelihood of the encoder's output under a fixed, pre-trained reference teacher model $s_{\text{real}}$. This corresponds to minimize real model' flow matching loss. Depending on the stop-gradient placement, we identify three variations, illustrated in Figure 3 (d, e, f). As the gradient update pushes latent codes $z$ solely towards high-density regions of $p_r(z)$, this objective lacks a mechanism to encourage entropy or coverage. Consequently, it suffers from **mode dropping**, where the latent distribution collapses into a few modes rather than covering the entire manifold.

**Fake Score Maximization (End-to-End).** This approach jointly trains the encoder and the diffusion model (student $s_{\text{fake}}$) without an explicit reference anchor. It aims to minimize the denoising error of the student model on its own generated samples. Similar to the failure mode discussed in REPA-E (Leng et al., 2025), this creates a self-reinforcing loop leads to mode collapse, which is more severe than real score maximization. As shown in Figure 3 (g, h, i), the trivial solution for minimizing this objective is for the latent space to contract to a single or a few points, which is trivial to model but useless for generation.

**Score Differences minimization.** An intuitive idea is to minimize the $L_2$ distance between score functions $||s_{\text{fake}}(z) - s_{\text{real}}(z)||^2$ thereby optimizing the input distribution. While theoretically sound, our experiments show that optimization is unstable, failing to align the distributions effectively as shown in Figure 3 (j, k, l).

**Loss Differences minimization.** This objective minimizes the difference in flow matching losses: $\mathcal{L} = ||v_{\text{real}} - \text{sg}(\epsilon - z)||^2 - ||v_{\text{fake}} - \text{sg}(\epsilon - z)||^2$. While it successfully aligns the distribution, the gradient computation involves the *Jacobian* of the score network, which is computationally expensive and memory-intensive for high-dimensional image latent spaces.

**Ours: Distribution Matching objective.** The distribution matching objective overcomes the aforementioned limitations. The gradient is approximated as the difference between score functions: $\nabla_z \mathcal{L}_{\text{DM}} \propto s_{\text{fake}}(z) - s_{\text{real}}(z)$. Crucially, this objective constructs a *difference vector field* that pulls them towards the high-density modes of $p_r(z)$ (via $+s_{\text{real}}$) and avoids mode collapse to a single mode through $-s_{\text{fake}}$. As illustrated in Figure 3 (b), this balanced dynamic achieves precise distribution alignment that preserves the global structure of the reference without the Jacobian overhead.

# 4. Experiments

Our core hypothesis is that the geometric structure of the latent space is critical for the efficiency and quality of the subsequent diffusion modelling. DMVAE provides a framework to systematically investigate this by explicitly controlling the aggregate posterior $q(z)$ to match a chosen reference distribution $p_r(z)$.

We conduct all experiments on the ImageNet $256\times256$ dataset. This section is organized as follows: We first investigate the properties of different reference distributions to identify the optimal prior for diffusion modeling. Building on this selection, we perform ablation studies to validate our design choices and hyperparameter settings. Finally, we conduct large-scale experiments to benchmark the optimized DMVAE against state-of-the-art methods.

## 4.1. Selection of Reference Distribution

The first and most critical question is: what is the optimal structure for a latent space to best support diffusion modeling? We use DMVAE as a probe to answer this. We test two categories of reference distributions, $p_r(z)$:

**Category 1: Data-Derived Distributions.** These priors are derived directly from the ImageNet training data, and are therefore semantically aligned with the data manifold.

- **SSL Features:** The aggregate posterior of a pre-trained DINOv2-ViT/L (Oquab et al., 2023) model.
- **Supervised Features:** The aggregate posterior of a supervised ResNet-34 (He et al., 2016) classifier.
- **Text Features:** We generate captions for ImageNet and extract features using a pre-trained SigLIP (Zhai et al., 2023) text encoder.
- **Diffusion Noise States:** The aggregate distribution of noisy latents $z_t$ (at $t = 0.5$) from a pre-trained LDM (Rombach et al., 2022) on ImageNet.

**Category 2: Data-Independent Distributions.** These priors are either synthetic or are not representative of the full data distribution.

- **Sub-sampled SSL features:** DINO features extracted from only 10 classes of ImageNet, creating a multi-modal distribution simpler than the full prior.
- **Standard Gaussian:** A simple $\mathcal{N}(0, I)$, as used in classic VAEs.
- **Gaussian Mixture Model (GMM):** A 10-component GMM, representing a simple, multi-modal synthetic baseline.

**Experimental Setting.** A fair comparison is challenging, as the difficulty of matching $q(z)$ to $p_r(z)$ depends on the distance and structural similarity between $p_r(z)$ and the original data manifold. As a preliminary setting for this exploration, we adopt a heuristic-based weighting. For data-

*Table 1.* Variations for different reference distributions.

| Ref distribution | rFID ↓ | PSNR ↑ | gFID-5k ↓ |
|---|---|---|---|
| VAE-baseline | 0.54 | 25.7 | 27.3 |
| DINO | 0.81 | 21.8 | 13.1 |
| Resnet | 1.46 | 20.9 | 18.6 |
| SigLIP-text | 1.63 | 24.0 | 26.8 |
| Difftraj | 0.60 | 26.9 | 31.8 |
| SubDino | 0.29 | 25.6 | 37.9 |
| GMM | 0.42 | 27.3 | 29.6 |
| Gaussian | 0.47 | 27.4 | 26.6 |

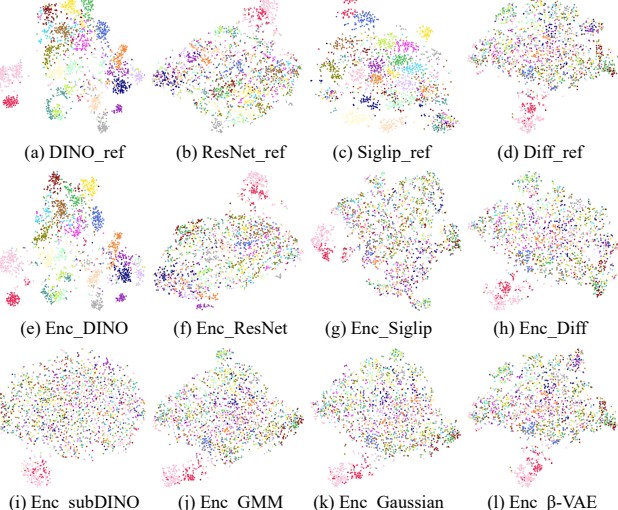

(a) DINO_ref  (b) ResNet_ref  (c) Siglip_ref  (d) Diff_ref

(e) Enc_DINO  (f) Enc_ResNet  (g) Enc_Siglip  (h) Enc_Diff

(i) Enc_subDINO  (j) Enc_GMM  (k) Enc_Gaussian  (l) Enc_β-VAE

*Figure 4.* Illustration of t-SNE on different distributions. (a-d) represent four different reference distributions, (e-h) represent the distribution of the DMVAE encoder learned from these four reference distributions, (i-k) represent the encoder distribution learned from the data-independent distribution, and (l) represents the distribution of the $\beta$-VAE encoder output.

derived priors, which are presumably closer to the original data, we set a higher matching weight of $\lambda_{\text{DM}} = 10$. For synthetic priors, we set $\lambda_{\text{DM}} = 1$. All other loss weights are held constant. All models are trained for 300k iterations with batch size 256.

The results, summarized in Table 1, show that the self-supervised DINO features provide the best overall balance of reconstruction quality and generative modeling performance. In addition, the comparison between the per-sample constraint (VAE baseline) and the distribution-level constraint (Gaussian) further demonstrates the superiority of distributional constraints.

**t-SNE Visualization and Analysis.** To understand *why* the SSL prior yields superior results, we performed a t-SNE visualization. We randomly sampled 100 images from 20 distinct ImageNet classes. We then visualized the reference distribution $p_r(z)$ and the latents $q(z)$ from our DMVAE trained to match reference distribution. The results are shown in Figure 4. The visualization clearly shows that

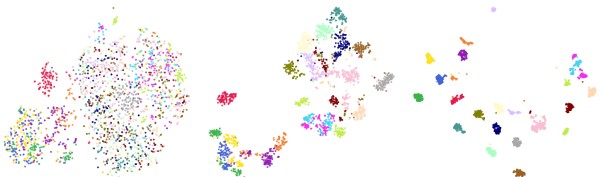

(a) cfg=1      (b) cfg=3      (c) cfg=10

*Figure 5.* Different CFG scale represents different distributions.

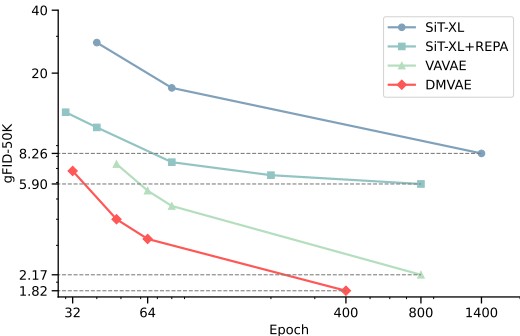

*Figure 6.* Comparison of the convergence speed.

the original DINO features ( Figure 4 a) possess superior semantic clustering. This inherent semantic organization likely reduces the complexity for the subsequent diffusion modelling, as it can focus on modeling intra-class variations rather than struggling with semantic confusion. Furthermore, the DMVAE latents ( Figure 4 e) successfully replicate this strong clustering structure, demonstrating that the distribution matching constraint effectively enforces the global geometry of the reference prior. In contrast, the $\beta$-VAE latents ( Figure 4 l) or DMVAE constrained by data-independent distributions demonstrated in the last row, struggles on distinguish different semantic modes. This confirms a more structured and semantically meaningful latent space is crucial for efficient generation.

### 4.2. Ablation Studies

We conduct ablation studies to validate the key design choices and hyper-parameters of DMVAE. Our default configuration, used as the baseline, employs the DINO reference distribution identified in Section 4.1, with $\lambda_{DM} = 10$ and a "Lightning-DiT-L" score network. We train each tokenizer for 250K iterations and the diffusion model for 300K iterations. We vary each factor independently, with results reported in Table 2.

- **DM Weight ($\lambda_{DM}$):** A weight of 10 provides the best balance. A lower weight (e.g., 1) results in poor generative quality due to weak regularization. Conversely, a higher weight (e.g., 100) strengthens the generative prior but degrades reconstruction performance by enforcing an overly strict constraint.
- **Score Network Size:** We observe that a larger score network, which can more accurately model the reference and latent distribution, leads to improved distribution matching and better final generative quality.
- **Classifier-Free Guidance (CFG):** While our default setting disables CFG (guidance weight of 1.0), we find that applying a small CFG weight (e.g., 3.0) during the score matching process (Equation (8)) can slightly improve generative quality, albeit at a minor cost to reconstruction. We visualize the how the real diffusion models the reference distribution under different CFG weights in Figure 5. We find that a suitable CFG brings a stronger semantic clustering properties, which can better accelerate convergence. When CFG scale is extremely large, the distribution may

be disjoint that makes training unstable.

- **Timestep schedule:** We compare our default uniform sampling ($t \sim U[0, 1]$) with a noise schedule annealing strategy. Following (Wang et al., 2023), the annealing setting starts at $[0, 1]$ and linearly contracts the sampling range to $[0, 0.5]$. We find that annealing provides a slight improvement by focusing on more precise, low-noise matching in the later stages of training.

### 4.3. Comparison with other methods

Finally, we compare our DMVAE against other state-of-the-art methods. The results are presented in Table 3. Our key finding is the exceptional training efficiency enabled by our approach. With a training budget of only 64 epochs, DMVAE achieves a gFID of 3.22. This result is highly competitive, outperforming other efficient methods like AlignTok (gFID 3.71) trained for the same duration. This remarkable efficiency is a direct consequence of using distribution-level constraint with DINO prior, which simplifies the modeling task for the diffusion network, as shown by the rapid convergence in Figure 6. When trained with larger budgets, DMVAE remains state-of-the-art. At 400 epochs, it surpasses most prior work, and at 800 epochs, it achieves a final gFID of 1.64, demonstrating the strong scalability and overall performance of our framework.

## 5. Related Work

Here, we discuss related work on the Visual Tokenizers and Distribution Matching. We present a more detailed related work discussion in Section A.

**Visual Tokenizers** Visual tokenizers underpin the "compress then model" paradigm by mapping images into compact discrete (Van Den Oord et al., 2017; Esser et al., 2021) or continuous (Rombach et al., 2022; Kingma & Welling, 2013) latent spaces. To address the trade-off between reconstruction quality and generative capability (Yao et al., 2025; Yu et al., 2023), recent works like VA-VAE (Yao et al., 2025) and AlignTok (Chen et al., 2025a) propose aligning latent spaces with semantic features from pre-trained Vision Foun-

*Table 2.* Ablation studies on diffusion matching weight, network size, CFG weight and timestep schedule.

| Factor | Setting | PSNR $\uparrow$ | rFID $\downarrow$ | gFID-5k $\downarrow$ |
|---|---|---|---|---|
| Default | ($\lambda = 10$, L-Net, CFG=1.0, Uniform) | 22.1 | 0.44 | 13.1 |
| DM Weight | $\lambda_{DM} = 1$ | 25.2 | 0.37 | 16.7 |
| | $\lambda_{DM} = 20$ | 21.4 | 0.78 | 12.5 |
| | $\lambda_{DM} = 100$ | 19.6 | 0.82 | 12.6 |
| Score Net Size | Small | 22.9 | 0.67 | 13.9 |
| | XL | 21.7 | 0.91 | 12.5 |
| CFG Weight | 3.0 | 20.6 | 1.02 | 11.6 |
| | 5.0 | 19.4 | 1.26 | 11.5 |
| Timestep Schedule | [0,1] annealing to [0,0.5] | 21.9 | 0.55 | 12.7 |

*Table 3.* Reconstruction and class-conditional generation results (without CFG) on ImageNet 256×256.

| Method | Reconstruction | | Training Epochs | #Params | Generation | |
|---|---|---|---|---|---|---|
| | PSNR$\uparrow$ | rFID$\downarrow$ | | | gFID$\downarrow$ | IS$\uparrow$ |
| *AutoRegressive (AR)* | | | | | | |
| LlamaGen (Sun et al., 2024) | 24.4 | 0.59 | 300 | 3.1B | 9.38 | 112.9 |
| MAR (Li et al., 2024) | 24.0 | 0.87 | 800 | 945M | 2.35 | 227.8 |
| *Latent Diffusion Models* | | | | | | |
| SiT (Ma et al., 2024) | 26.0 | 0.61 | 1400 | 675M | 8.61 | 131.7 |
| FasterDit (Yao et al., 2024) | 26.0 | 0.61 | 400 | 675M | 7.91 | 131.3 |
| RAE(DiT-XL) (Zheng et al., 2025) | 18.9 | 0.57 | 800 | 675M | 1.87 | 209.7 |
| VA-VAE (Yao et al., 2025) | 27.6 | 0.28 | 64 | 675M | 5.14 | 130.2 |
| VA-VAE (Yao et al., 2025) | 27.6 | 0.28 | 800 | 675M | 2.17 | 205.6 |
| AlignTok (Chen et al., 2025a) | 25.8 | 0.26 | 64 | 675M | 3.71 | 148.9 |
| AlignTok (Chen et al., 2025a) | 25.8 | 0.26 | 800 | 675M | 2.04 | 206.2 |
| DMVAE | 21.5 | 0.64 | 64 | 675M | 3.22 | 171.7 |
| DMVAE | 21.5 | 0.64 | 400 | 675M | 1.82 | 206.9 |
| DMVAE | 21.5 | 0.64 | 800 | 675M | 1.64 | 216.3 |

dation Models (VFMs) (Oquab et al., 2023; He et al., 2022; Radford et al., 2021; Zhai et al., 2023). While some approaches utilize frozen VFM encoders (Zheng et al., 2025), they often compromise fine-grained reconstruction details. Other efforts improve the autoencoder architecture or training recipe, such as DC-AE (Chen et al., 2025b;c) for high-compression-ratio tokenizers, LDMAE (Lee et al., 2025) for masked-autoencoder-based tokenizers, and Emu (Dai et al., 2023) for quality-tuning with curated data. Unlike these methods that focus on architecture design, data curation, or point-wise feature alignment, we propose shaping the global latent structure via distribution matching to achieve a more generative-friendly space.

**Distribution Matching** Aligning probability distributions is fundamental to generative modeling, exemplified by VAEs (Kingma & Welling, 2013; Higgins et al., 2017) and Adversarial Autoencoders (AAEs) (Makhzani et al., 2015). While AAEs introduce adversarial learning to match priors, they often struggle with complex target distributions. Our approach is most closely related to Variational Score Matching (Wang et al., 2023), a technique widely used in diffusion model distillation (Yin et al., 2024b; Zhou et al., 2024; Yin

et al., 2024a). We adapt this distribution-level regularization to align the visual tokenizer's latent distribution with a predefined prior, ensuring a better-structured latent space for subsequent modeling.

## 6. Conclusion and Limitation

In this work, we introduced the Distribution Matching VAE (DMVAE), a framework that, for the first time, allows the aggregate posterior of an autoencoder to be constrained at the *distributional level*. This enabled us to systematically explore the core question of "what kind of latent distribution is more beneficial for modeling." Our key finding is that using semantically-rich, self-supervised features like DINO as a reference distribution performs excellently in terms of both reconstruction fidelity and generative tractability, proving the superiority of this type of distribution. This approach is highly efficient, achieving an excellent gFID of 3.22 on ImageNet after only 64 epochs of training. We believe DMVAE offers a significant advancement for all two-stage generative models and can be broadly applied to tasks in audio, video, and 3D generation.

While our method is powerful, we acknowledge that matching distributions that are initially far apart remains a challenge, often requiring careful tuning. This makes the reference distribution more like a regularizer than a complete matching, which limits our precise control over the tokenizer's output distribution. In the future, we would focus on developing more robust optimization techniques for this distant-matching problem. Solving this will fully unlock the potential of designing generative latent spaces.

## Acknowledgements

LW is supported by National Science Foundation of China (NSFC92470123, NSFC62276005) and the State Key Laboratory of General Artificial Intelligence.

## Impact Statement

This paper presents work whose goal is to advance the field of Visual Generative Models. There are many potential societal consequences of our work, none of which we feel must be specifically highlighted here.

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

# A. Extended Related Work

**Visual Tokenizers**    Visual Tokenizers aim to compress high-dimensional image data into a compact latent representation, a strategy that has become fundamental to modern visual generative models (the "compress then model" paradigm). (Van Den Oord et al., 2017) pioneered this by introducing a Vector-Quantized Variational Autoencoder, which enabled autoregressive modeling in a discrete latent space, successfully moving beyond direct pixel-space operations. Building on this, VQGAN (Esser et al., 2021) further improved performance by integrating an adversarial loss into the VAE training objective, which significantly enhanced the perceptual quality and synthesis of fine details in reconstructed images. Concurrently, Latent Diffusion Models (Rombach et al., 2022) adapted this paradigm by training a VAE (Kingma & Welling, 2013) with a modest KL regularization, and subsequently modeling the resulting continuous latent distribution using a diffusion process. Recently, researches (Yao et al., 2025; Yu et al., 2023) has highlighted the reconstruction-generation dilemma, noting that improved reconstruction quality does not always correlate with superior generative performance, suggesting a potential trade-off (Yao et al., 2025). This observation has spurred research into alternative regularization methods. To mitigate this, approaches like VA-VAE (Yao et al., 2025) and AlignTok (Chen et al., 2025a) propose aligning the VAE's latent representations with the semantic features extracted from powerful, pre-trained Vision Foundation Models (VFMs) (Oquab et al., 2023; He et al., 2022; Radford et al., 2021; Zhai et al., 2023), EQ-AE (Kouzelis et al., 2025) proposes an Equivariance regularization. Furthermore, RAE (Zheng et al., 2025) explored using a frozen VFM as the encoder, training only a corresponding decoder for reconstruction. However, this strategy often results in poor reconstruction quality, suffering from a significant loss of fine-grained visual details. Beyond feature-alignment approaches, another line of work improves the autoencoder architecture and training recipe. DC-AE (Chen et al., 2025b) achieves spatial compression ratios of up to $128\times$ while maintaining reconstruction quality, and DC-AE 1.5 (Chen et al., 2025c) further introduces a structured latent space to accelerate diffusion convergence. LDMAE (Lee et al., 2025) analyzes desirable autoencoder properties for latent diffusion and proposes masked autoencoders as an alternative backbone. Emu (Dai et al., 2023) demonstrates that quality-tuning with curated high-quality data can boost generation fidelity. While these methods advance autoencoder architecture, compression efficiency, or data curation, they all leave the aggregate latent distribution unregulated. Unlike these methods, we expand the concept of regularization to distribution matching and directly shape the entire latent distribution to a desired prior, leading to a more globally consistent and generative-friendly latent space.

**Distribution Matching**    Distribution matching is a long-standing problem in generative modeling, focusing on finding a transformation or mapping (transport) between different probability distributions. Variational Autoencoders (VAEs) (Kingma & Welling, 2013; Higgins et al., 2017) apply this by aiming to match the encoded hidden states of images to a simple Gaussian distribution, which then enables image generation from randomly sampled Gaussian vectors. To mitigate the restrictiveness of the standard Gaussian prior, several approaches focus on increasing the expressivity of the prior distribution to better fit the aggregated posterior (Tomczak & Welling, 2018; Bauer & Mnih, 2019; Kuzina & Tomczak, 2024). Adversarial Autoencoders (AAE) (Makhzani et al., 2015) addressed the common mode collapse issue (Rezende & Viola, 2018; Razavi et al., 2019) of VAEs by introducing adversarial learning (Goodfellow et al., 2020) to enforce the alignment of the image latent distribution with an arbitrary prior. However, AAE struggles to match complex target distributions due to the limited capacity of the standard discriminator. The most related technique to our work is the Variational Score Matching (Wang et al., 2023) used in diffusion model distillation (Wang et al., 2023; Yin et al., 2024b; Zhou et al., 2024; Yin et al., 2024a). This technique successfully distills a student generative model from a teacher by directly comparing and aligning their respective score functions. We adopt this distribution matching technique, but apply it to match the hidden representations of visual tokenizers with a predefined prior distribution. This distribution-level regularization allows us to shape a better-structured and more generative latent space for subsequent image modeling.

# B. Implementation Details

Our methodology involves a multi-stage training pipeline. We detail the hyperparameters and architecture for each stage below.

## B.1. Tokenizer Pretraining

We first train an initial AutoEncoder compressing the latents to a low dimension. We employ the pre-trained DINO-v2-large model (300M parameters) as our visual encoder. Given an input resolution of $256 \times 256$, the encoder yields a $16 \times 16$ latent feature map. Following (Chen et al., 2025a), we introduce an MLP projection head with a hidden dimension of 2048 to project the encoder features into a compact latent dimension of 32. For the decoder, we adopt a convolution-based

architecture similar to Flux (Labs, 2024). During this phase, we freeze the parameters of the DINO-v2 encoder and exclusively optimize the MLP projector and the decoder. The model is trained on the ImageNet dataset with a batch size of 256 and a learning rate of $1 \times 10^{-4}$ for 8 epochs (40k steps).

We optimize the tokenizer using a composite objective function comprising reconstruction and adversarial terms. Following (Esser et al., 2021), the total loss $L_{\text{total}}$ is:

$$L_{\text{total}} = L_1 + L_{\text{perceptual}} + \lambda_{\text{gan}} L_{\text{gan}} \tag{10}$$

where we set the adversarial weight $\lambda_{\text{gan}} = 0.5$.

### B.2. Reference Distribution (Teacher) Pretraining

In this stage, we train a teacher model to capture the reference distribution. We first extract the latent representations for the entire ImageNet dataset using the frozen encoder and projector from the previous stage trained tokenizer. We then employ the LightningDiT-XL architecture with QKNorm (675M parameters) as our generative backbone to model these latents. This teacher model is trained to minimize the flow matching loss (Eq. 3). We follow the standard training recipe from VAVAE(Yao et al., 2025), using a batch size of 1024, a learning rate of $2 \times 10^{-4}$, and train for 400 epochs (500K steps).

### B.3. Distribution Matching VAE Training

This stage is the core of our method, where we jointly train the VAE and a student model to align the VAE's latent space with the reference distribution. The framework involves three components: the AE (from Tokenizer pretraining stage), the teacher model (from reference pretraining stage) and the student model (initialized from teacher model).

We initialize the AE from the tokenizer pretraining stage checkpoint and the student model with the weights of the frozen teacher. The optimization is twofold:

- **VAE:** The VAE is updated by minimizing a composite objective: the reconstruction loss (Eq. 10) and the distribution matching (DM) loss (Eq. 8). The DM loss serves to align the VAE's latent posterior with the reference distribution defined by the teacher.

- **Student Model:** The student ("fake") model is simultaneously optimized to model the VAE's evolving latent distribution by minimizing a score-based loss (Eq. 7).

We adopt a learning rate of $2 \times 10^{-5}$ for both trainable models (the VAE and the student). The distribution matching loss term is scaled by a weight of $\lambda_{\text{dm}} = 10$, and we apply a classifier-free guidance (CFG) scale of $w = 5$ during the computation of the DM loss. Following the strategy in (Yin et al., 2024a), we employ an alternating update schedule: the VAE is updated once every 5 iterations, while the student model is optimized at every training step. The joint training process is conducted for 350K steps.

### B.4. Decoder Fine-tuning

Similar to (Chen et al., 2025a), we fine-tune the decoder of the DM-VAE. We freeze the encoder and MLP projector, optimizing only the decoder with the reconstruction objective (Eq. 10). This fine-tuning is run for 250K steps, using a batch size of 256 and a learning rate of $1 \times 10^{-4}$.

### B.5. Diffusion Model Training

After the DM-VAE is fully trained, we now train a final, high-fidelity generative model within its latent space. We train a new LightningDiT-XL model from scratch on DM-VAE latents, again optimizing for the flow matching loss (Eq. 3). This training is conducted with a batch size of 1024, a learning rate of $2 \times 10^{-4}$, and runs for 800 epochs (1M steps).

## C. Visualization of DMD Gradient Direction

We visualize the DMD update vectors (i.e., the negative gradient field) using a 2D toy example in Figure 7. As observed, relying on a large noise scale leads to coarse and potentially inaccurate update directions. However, high noise levels are

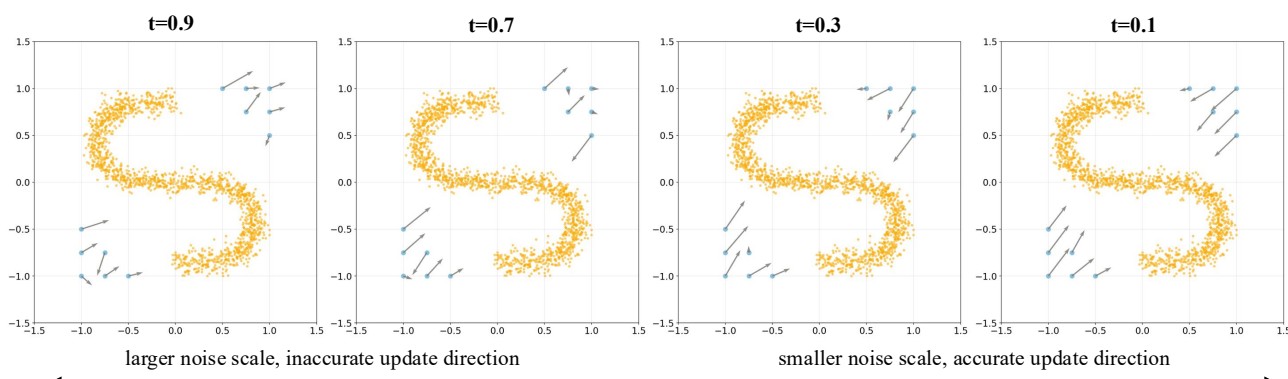

*Figure 7.* DMD update direction visualized at different timesteps in a 2D toy setting.

practically necessary to bridge the gap between distributions with initially disjoint supports. Conversely, a small noise scale yields highly precise update directions, but may fail to provide meaningful guidance when the distributions are far apart. The training process can be regarded as an averaging of different time steps, which demonstrates the value of dynamically adjusting the noise schedule during training. This needs further exploration.

## D. Stabilizing the Distant Distribution Matching.

We empirically found that the joint-training objective (Equation (9)) can be unstable when the aggregate posterior $q(z)$ is far from the reference $p_r(z)$. This stems from a fundamental dilemma in score matching (Song et al., 2020; Yin et al., 2024b).

- On one hand, using a **large noise scale** $\sigma_t$ ensures support overlap, but this level of smoothing "washes out" the fine-grained, high-frequency differences between the distributions, leading to a flat loss landscape for $\mathcal{L}_{\text{DM}}$ and causing vanishing gradients.

- On the other hand, a **small noise scale** $\sigma_t$ preserves distribution details but, when $q(z)$ and $p_r(z)$ is distant, it suffers from disjoint supports. This causes high-variance, explosive gradients that destabilize the entire joint-training process and can lead to mode collapse.

Crucially, unlike prior works (Huang et al., 2025; Yin et al., 2024b) leveraging distribution matching for distillation—where an implicit pairing or regression loss keeps the distributions proximate—our framework is, to our knowledge, the first to use this mechanism to match the entire aggregate posterior $q(z)$ of an autoencoder to an unpaired reference $p_r(z)$. The potentially vast initial discrepancy $D(q(z)||p_r(z))$ makes the system highly susceptible to the aforementioned instabilities.

To effectively mitigate these inherent training difficulties and ensure convergence, we adopts several stablizing training strategies: we utilize the pretrained weights of the real score model to initialize the fake score model and train the VAE from a pretrained encoder, providing a stable starting point; implement alternating training between the VAE and the fake score model to balance the learning dynamics (Yin et al., 2024b). Furthermore, we reduce all latent spaces and priors to a low dimension (e.g., $d = 32$) with reconstruction objective. This step is critical as it helps mitigate the curse of dimensionality, thus reducing the sparsity and computational challenges associated with distribution matching in high-dimensional spaces.

## E. Convergence Speed Visualization

We present a visual comparison of the convergence speed between DMVAE and VAVAE in Figure 6. The results demonstrate that DMVAE achieves consistently superior visual generation quality compared to VAVAE when evaluated at equivalent training steps.

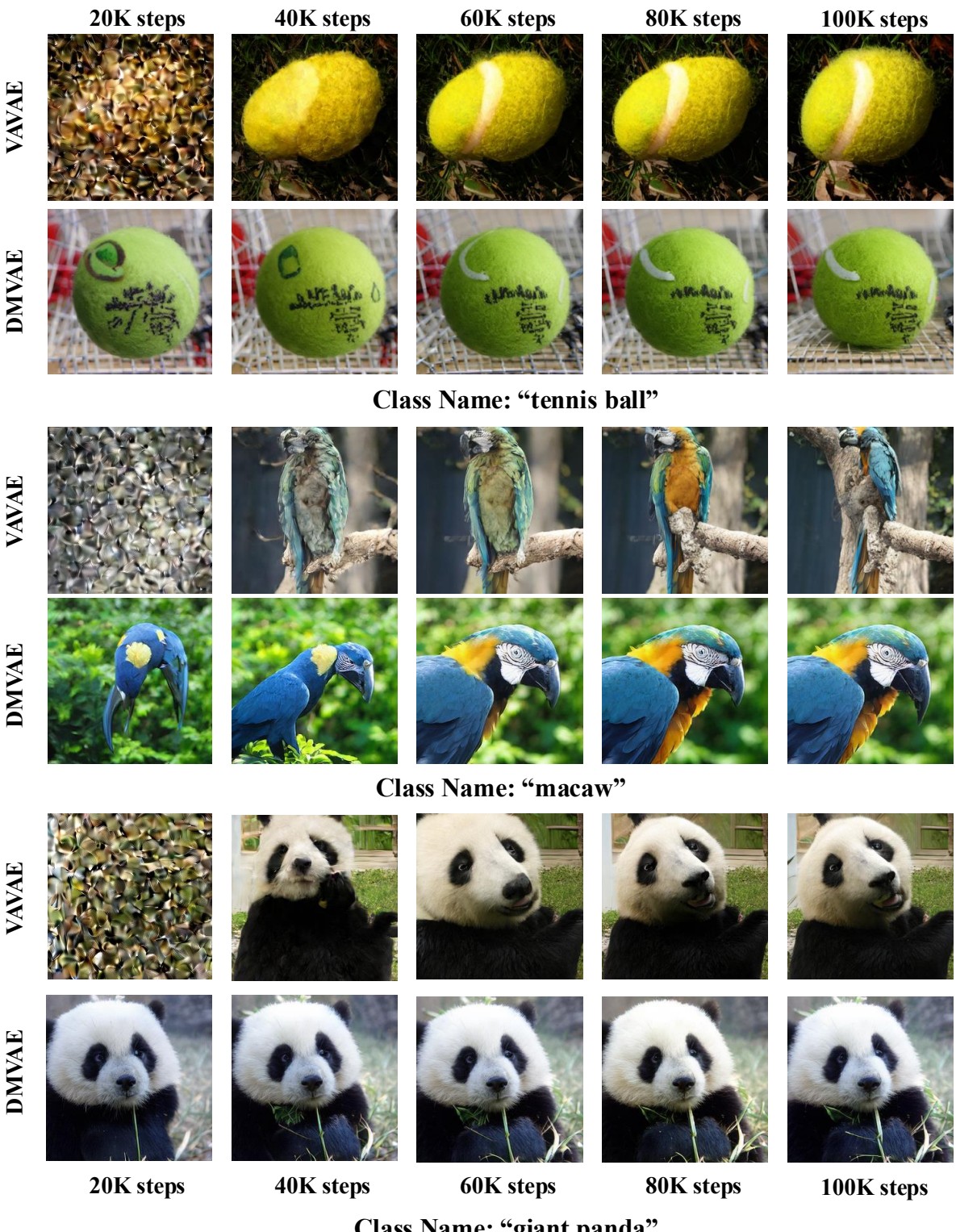

*Figure 8.* **Quanlitative comparison of convergence speed on ImageNet 256×256.** We compare DMVAE with VAVAE and report conditional results without CFG.

