# OpenReview forum: "Distribution Matching Variational AutoEncoder"
_ICML.cc/2026/Conference — ICML 2026 regular_

### Official Review · Reviewer_G7gD · 2026-03-04

**Soundness:** 3
**Presentation:** 3
**Significance:** 2
**Originality:** 3
**Overall Recommendation:** 4
**Confidence:** 4

**Summary:**

The paper proposes Distribution-Matching VAE (DMVAE), a visual tokenizer that explicitly shapes the aggregate latent distribution rather than relying on per-sample KL regularization as in standard VAEs or pointwise feature alignment. DMVAE matches the encoder’s aggregate posterior q(z) to an arbitrary reference distribution p_r(z) using a score-based distribution-matching constraint, implemented with a frozen teacher diffusion/flow score model for p_r and an online student score model tracking the evolving q(z). This enables controlled studies of latent priors. Experiments on ImageNet-256 show self-supervised (DINO) feature distributions yield latents that are easier for diffusion models to learn, achieving strong generation quality with a reconstruction–modeling trade-off.

**Compliance With Llm Reviewing Policy:**

Affirmed.

**Final Justification:**

After considering the rebuttal, I am comfortable raising my score from Weak Reject to Weak Accept. Details see comments.

**Key Questions For Authors:**

1. How well does DMVAE quantitatively match the target reference distribution p_r(z)? Please report one or more distribution-level metrics between samples from the learned aggregate posterior q(z) and p_r(z) (e.g., Sliced Wasserstein, MMD, precision/recall in latent space, or another justified divergence/score-based measure), and correlate them with downstream gFID.
2. Compute accounting: what is the end-to-end training cost compared to baselines? DMVAE involves multi-stage training (tokenizer pretraining, teacher diffusion for p_r, joint training with an online student, decoder finetuning, then the final diffusion model). Please provide a fair comparison that includes total GPU-hours / FLOPs for DMVAE vs. VAVAE/AlignTok/RAE under matched settings.
3. Hyperparameter fairness: how sensitive are conclusions to  \lambda_{DM} and other tuning across priors?In the prior-selection study, different priors use different \lambda_{DM} heuristics. Can you run a controlled sweep where each prior gets a comparable tuning budget (e.g., same grid or Bayesian search budget), and report whether DINO remains best?
4. What exactly improves diffusion learnability? When does distribution matching break, and what are the practical safeguards? Can you provide evidence that the gains come from properties like clustering/separability (e.g., class-conditional variance structure, intrinsic dimension, spectral statistics, or linear probe separability), rather than incidental factors (e.g., latent scaling, anisotropy, or teacher/model capacity)?

**Limitations:**

Not fully. Please refer to  the Weaknesses.

**Strengths And Weaknesses:**

Strengths
* Clear motivation and reasonable methodology. The paper targets a concrete gap in two-stage generative modeling, and proposes a technically plausible score-based distribution matching approach.
* The "systematic study of latent priors" angle is insightful. Using the proposed framework as a probe to compare different reference distributions (e.g., SSL vs. Gaussian/text/supervised features) helps turn a vague design choice into an empirical question.
* A meaningful conceptual step beyond pointwise feature alignment. By transferring DMD/score distillation ideas to aggregate posterior matching, the approach aims to control distribution-level geometry rather than only enforcing sample-wise alignment (as in many VFM-alignment tokenizers).
* Well written with strong visual support. The narrative is easy to follow, and figures/tables (e.g., t-SNE and convergence plots) effectively communicate the empirical story.

Weaknesses
* The "match to an arbitrary distribution" claim may be overstated. In practice, stable optimization when q(z) and p_r(z) are far apart appears challenging and may require careful tuning, suggesting the method can behave more like a strong regularizer than a mechanism for precise, general-purpose distribution control.
* The heuristic choice of different \lambda_{DM} values across priors (e.g., larger for data-derived priors and smaller for synthetic ones) may affect the conclusion that "DINO is optimal." A more systematic comparison under matched budgets and comparable hyperparameter search spaces would strengthen robustness.
* The reconstruction-generation trade-off remains somewhat empirical. For example, the SSL/DINO prior improves gFID while significantly reducing PSNR; the paper attributes this to easier diffusion modeling, but offers limited analysis of what information is lost (frequency vs. semantics, texture vs. structure) and whether better Pareto trade-offs are achievable (e.g., hierarchical latents, local–global factorization, or multi-scale objectives).

---

> ### Author Rebuttal · Authors · 2026-03-30
>
> We thank Reviewer G7gD for recognizing our clear motivation, systematic study of latent priors, meaningful step beyond pointwise alignment, and strong visual support. We address your concerns below:
>
> ---
>
> **Q1: Quantitative distribution matching metrics and correlation with gFID.**
>
> We report MMD and SWD between $q(\mathbf{z})$ and the DINO reference across all ablation settings:
>
> | Setting | gFID-5k $\downarrow$ | MMD $\downarrow$ | SWD $\downarrow$ |
> |---|---|---|---|
> | $\lambda_{DM}$ = 1 | 16.7 | 0.243 | 1.212 |
> | $\lambda_{DM}$ = 10 (default) | 13.1 | 0.055 | 0.506 |
> | $\lambda_{DM}$ = 20 | 12.5 | 0.026 | 0.421 |
> | $\lambda_{DM}$ = 100 | 12.6 | 0.014 | 0.320 |
> | Score Net: Small | 13.9 | 0.081 | 0.700 |
> | CFG = 3.0 | 11.6 | 0.022 | 0.395 |
> | CFG = 5.0 | 11.5 | 0.017 | 0.351 |
>
> Strong monotonic relationship: as $\lambda_{DM}$ increases, MMD/SWD decrease and gFID improves. The slight gFID increase at $\lambda=100$ reflects the reconstruction--matching trade-off (PSNR: 21.4→19.6). VAE-baseline vs. Gaussian DMVAE (same $\mathcal{N}(0,I)$ target, same architecture) further validates the mechanism:
>
> | Method | MMD | SWD | gFID |
> |---|---|---|---|
> | VAE-baseline (per-sample) | 0.09 | 1.06 | 27.3 |
> | Gaussian DMVAE (dist-level) | 0.07 | 0.81 | 26.6 |
>
> **Q2: End-to-end training cost.**
>
> | Stage | VA-VAE | DMVAE |
> |---|---|---|
> | Tokenizer | $1.18 \times 10^{20}$ | $6.80 \times 10^{20}$ |
> | Final diffusion | $9.01 \times 10^{20}$ (800ep) | $4.50 \times 10^{20}$ (400ep) |
> | **Total** | **$1.02 \times 10^{21}$** | **$1.11 \times 10^{21}$** |
> | **gFID** | **2.17** | **1.82** |
>
> ~9% more FLOPs, 16% better gFID. We clarify that our final diffusion model is trained **from scratch** on the finalized latent space --- it does not continue from the teacher/student models. The convergence speed comparison (Figure 6) measures how quickly a fresh DiT learns the latent distribution, purely reflecting latent space quality. Including teacher cost would conflate tokenizer preparation with latent learnability. If our goal were purely to optimize gFID, we could continue from the teacher/student checkpoint --- but this would not demonstrate the latent space's intrinsic learnability. The tokenizer is trained once and reused for all downstream diffusion training. Student overhead: ~549 GFLOPs/step average, ~4 GB memory (<10%), no backward through student.
>
> **Q3: $\lambda_{DM}$ fairness across priors.**
>
> Even at its *weakest* ($\lambda=1$, gFID 16.7), DINO outperforms all non-SSL priors at their tuned $\lambda$: ResNet (18.6), SigLIP (26.8), Difftraj (31.8), Gaussian (26.6), GMM (29.6). No reasonable $\lambda_{DM}$ reverses the ranking. Data-derived priors have richer score gradients and tolerate larger $\lambda_{DM}$ without reconstruction collapse, while synthetic priors require smaller $\lambda_{DM}$ --- a principled scaling of regularization to signal complexity. We will include additional sweep data in the revision.
>
> **Q4: What improves diffusion learnability?**
>
> Five lines of evidence: **(a)** MMD/SWD--gFID correlation across 7 settings establishes distributional alignment itself drives improvement. **(b)** Score network size ablation (13.9→13.1→12.5) rules out teacher capacity as dominant factor. **(c)** $\lambda_{DM}$ ablation varies only matching strength with fixed architecture/normalization, ruling out incidental scaling. **(d)** Figure 4: DINO reference (4a) shows semantic clustering, DMVAE (4e) replicates it, while $\beta$-VAE (4l) and data-independent priors (4j,k) show diffuse distributions --- aligning with gFID ranking. **(e)** The "deceptive minimization" analysis (Sec. 3) theoretically explains why distribution-level constraints produce better-structured latent spaces.
>
> **W1: "Arbitrary distribution" overstated.**
>
> We agree "arbitrary" is too strong, but DMVAE goes beyond mere regularization --- it performs **controllable distribution matching**. MMD decreases monotonically from 0.243 to 0.014 (17$\times$ reduction) as $\lambda_{DM}$ increases. We balance matching and reconstruction at $\lambda=10$--$20$. We revised to "explicitly align the latent distribution with a diverse family of reference distributions."
>
> **W3: Reconstruction-generation trade-off.**
>
> Table 2 traces the full Pareto curve: $\lambda=1$ (PSNR 25.2, gFID 16.7), $\lambda=10$ (22.1, 13.1), $\lambda=20$ (21.4, 12.5), $\lambda=100$ (19.6, 12.6). Diminishing gFID returns beyond $\lambda=20$ while PSNR continues to decrease identifies the optimal regime. DINO features emphasize semantic structure over high-frequency detail (Oquab et al., 2023); the PSNR reduction reflects loss of texture information that is hardest for diffusion to learn, while preserving semantics that drive generation quality. The faster convergence in Figure 6 confirms this. We agree that hierarchical latents or multi-scale objectives could further improve the Pareto trade-off and will discuss this in the revision.

---

> > ### Author Rebuttal · Reviewer_G7gD · 2026-04-03
> >
> > Thank you to the authors for the thoughtful rebuttal. The additional quantitative evidence substantially strengthens the paper’s main claim that DMVAE improves latent-space structure at the distribution level rather than merely acting as a heuristic regularizer. The added compute accounting is also helpful and makes the efficiency claim more credible. I also appreciate the authors’ clarification and moderation of the “arbitrary distribution” claim, which makes the scope of the contribution more precise. While I still think the fairness of hyperparameter tuning across priors and the mechanistic explanation could be further strengthened in the final version, the rebuttal addresses my main concerns sufficiently.
> >
> > I am therefore updating my score from Weak Reject to Weak Accept, and encourage the authors to add comprehensive evidence in the revised version if possible.

---

### Official Review · Reviewer_o3xD · 2026-03-10

**Soundness:** 3
**Presentation:** 3
**Significance:** 3
**Originality:** 3
**Overall Recommendation:** 4
**Confidence:** 5

**Summary:**

The paper introduces Distribution-Matching VAE (DMVAE), a novel framework designed to explicitly align the aggregate posterior of an autoencoder's latent space with a predefined reference distribution. This approach moves beyond the per-sample KL-divergence constraints of standard VAEs, which often fail to regulate the global geometry of the latent manifold.

**Compliance With Llm Reviewing Policy:**

Affirmed.

**Key Questions For Authors:**

How much additional computational overhead is introduced by the simultaneous training of the online student score model?

**Limitations:**

Yes

**Strengths And Weaknesses:**

###Strengths
1. Unlike conventional VAEs that rely on pointwise KL constraints—often resulting in a lack of manifold continuity—DMVAE introduces a novel paradigm that explicitly optimizes the global geometric structure of the latent space via score-based distribution matching.
2. The paper provides a comprehensive empirical analysis of various priors, including Gaussian, text embeddings, and SSL (e.g., DINO) features, effectively identifying the latent space characteristics that maximize generative model performance.

###Weaknesses
While the paper demonstrates impressive performance, the complexity of the methodology raises concerns regarding its practical robustness and whether the performance gains justify the overhead.

1. [Complex Training Pipeline] Achieving optimal results requires at least four distinct stages: tokenizer pre-training, teacher model training, joint training of the VAE and student model, and finally, decoder fine-tuning. This multi-step process risks accumulating errors at each stage and imposes a high "training fatigue" compared to the trend of end-to-end learning.
2. [High Computational Resource Requirements] To track the evolving distribution of the encoder in real-time, an "online student score model" must be updated simultaneously. This significantly increases memory usage and computational complexity compared to standard VAEs, potentially hindering scalability to larger models.
3. As noted in the text, if the initial latent distribution generated by the encoder is too far from the reference distribution (e.g., DINO), there is a significant risk of vanishing or exploding gradients during score matching, leading to mode collapse. The requirement for meticulous hyperparameter tuning and stabilization strategies to prevent this is a notable drawback for generalizability.

### Minor Issues

Typo: There is a reference error [Eq.??] at Line 632 that requires correction.

Missing References: Several key studies addressing AutoEncoders within the LDM framework are missing from the citations. In particular, the following papers regarding MAE-based latent learning or structured optimization are highly relevant:

[1] Latent Diffusion Models with Masked Autoencoders, Lee et al.
[2] Emu: Enhancing Image Generation Models Using Photogenic Needles in a Haystack, X. Dai et al.
[3] DC-AE 1.5: Accelerating Diffusion Model Convergence with Structured Latent Space, MIT Han Lab
[4] Deep Compression Autoencoder for Efficient High-Resolution Diffusion Models, MIT Han Lab

---

> ### Author Rebuttal · Authors · 2026-03-30
>
> We thank Reviewer o3xD for recognizing the novel paradigm of optimizing global geometric structure via score-based distribution matching and the comprehensive empirical analysis. We address your questions below:
>
> ---
>
> **W1: Complex multi-stage training pipeline.**
>
> We acknowledge the pipeline is more complex than standard tokenizers. Existing methods (e.g. AlignTok) use 3 stages; DMVAE adds 2: teacher pretraining (B.2) and joint DM training (B.3).
>
> **The complexity is in tokenizer training, which is a one-time cost.** Once trained, the tokenizer produces a fixed latent space reusable for multiple diffusion models with different architectures or recipes. The final diffusion training (the most expensive and frequently repeated stage) is identical to standard practice.
>
> **Errors do not accumulate.** B.3 reshapes B.1's distribution; B.4 corrects reconstruction degradation; B.5 trains from scratch on finalized latents. Teacher and student models are discarded after tokenizer training.
>
> **Four of five stages require no novel tuning.** B.1 follows AlignTok's tokenizer recipe; B.2 follows VA-VAE's flow matching recipe; B.4 follows AlignTok's decoder finetuning recipe; B.5 is standard DiT training. Only B.3 introduces new hyperparameters ($\lambda_{DM}$, alternating schedule), and Table 2 shows the method is robust across a wide range of $\lambda_{DM}$.
>
> We acknowledge pipeline simplification as a valuable future direction --- e.g., merging B.1 and B.3 into a single stage with warm-up scheduling, or using publicly available diffusion checkpoints for B.2 to eliminate a training stage entirely.
>
> ---
>
> **W2: Computational overhead from online student score model.**
>
> Concrete measurements during Stage B.3:
> - VAE: 2749 GFLOPs/update; Student DiT: 1131 GFLOPs/update
> - VAE updates every 5 iterations, student every iteration → average overhead **~549 GFLOPs/step**
> - Additional memory: **~4 GB (<10%** of peak usage)
> - DMD update does **not** backpropagate through the student --- only a forward pass
>
> Total pipeline comparison with VA-VAE:
>
> | Method | Total FLOPs | gFID |
> |---|---|---|
> | VA-VAE | $1.02 \times 10^{21}$ | 2.17 |
> | DMVAE | $1.11 \times 10^{21}$ | 1.82 |
>
> ~9% more total FLOPs, but substantially better gFID. DMVAE needs only 400 diffusion epochs to surpass VA-VAE's 800-epoch result. The extra tokenizer cost is offset by faster diffusion convergence, and the tokenizer is trained once and reused for all downstream diffusion training.
>
> **Q1:** Addressed in W2. In summary: ~549 GFLOPs/step average overhead, ~4 GB additional memory (<10%), no backward pass through the student.
>
> ---
>
> **W3: Risk of vanishing/exploding gradients when distributions are far apart.**
>
> **In the majority of experiments, this was not observed.** Across all data-derived priors (DINO, ResNet, Difftraj) and data-independent priors (Gaussian, GMM), training was stable with our standard strategies: pretrained initialization, alternating schedule (VAE updated every 5 iterations), and $d=32$ projection.
>
> **The only case requiring special handling was SigLIP text embeddings**, which are structurally very different from initial encoder output. We resolved this by pre-training a lightweight MLP aligner (a few thousand iterations) to reduce the initial discrepancy before DM training. This simple solution was only needed for this single case.
>
> ---
>
> **Typo at L632:** We have fixed it, with full-document cross-reference check.
>
> **Missing references:** We have added all four papers to related works and discuss their complementary relationship with DMVAE.

---

> > ### Author Rebuttal · Reviewer_o3xD · 2026-04-03
> >
> > I appreciate authors for their response. I will keep my positive score. I wish the authors the best with their work.

---

### Official Review · Reviewer_aRBa · 2026-03-14

**Soundness:** 3
**Presentation:** 3
**Significance:** 4
**Originality:** 2
**Overall Recommendation:** 5
**Confidence:** 4

**Summary:**

Problem statement
* The latent space of visual generative models should match a distribution which are optimal for modeling.

Proposed component: Distribution-Matching VAE (DMVAE)
* DMVAE explicitly aligns the encoder’s latent distribution with an arbitrary reference distribution
* reference distributions
    * Gaussians as conventional VAEs)
    * Empirical distribution for self-supervised features
    * Gaussians? for diffusion noise
    * others
* Reference distributions are represented by the score estimated by a diffusion model. This model is pre-trained and frozen.
* How to align the reference distribution and the encoded distribution?
    * The encoder produces a latent.
    * The latent is fed into the frozen reference diffusion model and a new diffusion model.
    * Compute the typical diffusion loss on the new diffusion model. It trains the new diffusion model.
    * Compute a distribution matching loss (= score matching) between the new and reference diffusion models. It trains the encoder.
    * Compute reconstruction loss through the encoder and the decoder, and train them.

Findings
* SSL-derived distributions are good for both reconstruction fidelity and modeling efficiency.
* reconstruction fidelity = reconstruction from the latents
* modeling efficiency = convergence speed and modeling quality = gFID 3.2 / 1.82 on ImageNet in 64 / 400 training epochs

**Compliance With Llm Reviewing Policy:**

Affirmed.

**Final Justification:**

I appreciate the authors for the clarification. Luckily, between the initial review and the rebuttal, I have read DMD and variants to get enough background knowledge. I hope the authors will put enough explanations or pointers for the readers not familiar with DMD such as the former myself.

Furthermore, the additional numbers regarding convergence speed and alignment resolved my concerns.

I increased my score to accept.

**Key Questions For Authors:**

Resolving weaknesses 1-6 may raise my score.

**Limitations:**

The authors mention only the limitation in the matching.

I think more critical limitation is theoretical or empirical supports.

**Strengths And Weaknesses:**

### Strengths

The background covers the recent literature.

Figure 1 is well presented to compare the approaches.

The training procedure of DMVAE is reasonably designed and compared to its alternatives in a toy experiment.

DMVAE has advantages of VAE and RAE.

### Weaknesses

 1. It is confusing to name the reference and new diffusion models as teacher and student models.
    * It implies the student model receives some kind of distillation loss from the teacher directly, but it follows the teacher passively as the encoder follows the distribution of the reference.

1. The convergence speed comparison should include pre-training the reference diffusion model to support efficiency.

1. I wonder how we can measure the alignment between the distribution of the latent of DMVAE and the reference distribution and how much they align.
1. How would we compute D_KL in Figure 3b when the reference distribution is given only as an empirical distribution? I suppose it is implicitly minimized by the training procedure of DMVAE. If the training procedure exactly minimizes D_KL, it should accompany a proof.

1. Figure 4 The t-SNE visualization does not guarantee the correct comparison of semantic clustering. I suggest to use quantitative measures instead.

1. The finding “SSL-derived distributions are good” is supported by a limited configuration which has only 10 classes of ImageNet. How would it work on the full dataset? Full ImageNet is already loose to expect the proposed method generalize to recent generative models.  Only 10 classes is too small.


1. Somewhat irrigorous statements and words
    * L022 a latent space is too complex and irregular - what is complex and irregular?
    * L030, L159 KL divergence operates on individual samples and ignores the aggregate distribution … - why does not the aggregated distribution follow Gaussian distribution even though individual samples follow Gaussian distribution? I agree to the possibility of collapsing to a delta function and the special case in L190, but the possibility is tiny.
    * modeling efficiency = convergence speed and modeling quality - it should be explained in the first occurence.

1. L_align should be defined.

1. The approach somewhat lacks novelty: the principle of matching the distributions with another encoder is the same with Pointwise Matching and the new thing is how to match.

1. (minor) Please use either modelling or modeling.

---

> ### Author Rebuttal · Authors · 2026-03-30
>
> We thank Reviewer aRBa for recognizing our well-presented background, reasonably designed training procedure, and the advantages of DMVAE over VAE and RAE. We address your questions below:
>
> ---
>
> **W1: "Teacher/student" terminology.**
>
> The terminology follows the DMD framework (Yin et al., 2024b). The student is not passive --- it estimates the score of the current  $q(z)$ , and the score difference ( $s_{\text{fake}} - s_{\text{real}}$ ) drives encoder updates via Eq. (8). We will add a clarifying remark in the revision.
>
> **W2: Convergence speed should include teacher pretraining cost.**
>
> We provide a complete FLOPs comparison with VA-VAE:
>
> | Stage | VA-VAE | DMVAE |
> |---|---|---|
> | Tokenizer (total) | $1.18 \times 10^{20}$ | $6.80 \times 10^{20}$ |
> | Final diffusion | $9.01 \times 10^{20}$ (800ep) | $4.50 \times 10^{20}$ (400ep) |
> | **Total** | **$1.02 \times 10^{21}$** | **$1.11 \times 10^{21}$** |
> | **gFID** | **2.17** | **1.82** |
>
> Under ~9% more FLOPs, DMVAE achieves substantially better gFID. Importantly, our final diffusion model is trained **from scratch** --- it does not continue from the teacher/student. The convergence speed in Figure 6 measures how quickly a fresh DiT learns the latent distribution, purely reflecting latent space quality. Including teacher cost conflates tokenizer preparation with latent learnability. The tokenizer is trained once and reused for all downstream diffusion training.
>
> **W3: How to measure alignment quality.**
>
> We report MMD and SWD between $q(\mathbf{z})$ and the DINO reference $p_r(\mathbf{z})$:
>
> | Setting | gFID-5k $\downarrow$ | MMD $\downarrow$ | SWD $\downarrow$ |
> |---|---|---|---|
> | $\lambda_{DM}$ = 1 | 16.7 | 0.243 | 1.212 |
> | $\lambda_{DM}$ = 10 (default) | 13.1 | 0.055 | 0.506 |
> | $\lambda_{DM}$ = 20 | 12.5 | 0.026 | 0.421 |
> | $\lambda_{DM}$ = 100 | 12.6 | 0.014 | 0.320 |
>
> Stronger matching (lower MMD/SWD) consistently correlates with better gFID. The slight gFID increase at $\lambda=100$ despite lower MMD/SWD reflects the reconstruction--matching trade-off (PSNR drops from 21.4 to 19.6), identifying the optimal regime at $\lambda=10$--$20$.
>
> **W4: $D_{\text{KL}}$ in Figure 3b.**
>
> The gradient in Eq. (8) corresponds to a weighted $D_{\text{KL}}(q_t \| p_{r,t})$ integral over noise levels (Wang et al., 2023; Yin et al., 2024b), driving $D_{\text{KL}}(q \| p_r)$ toward zero.
>
> **W5: t-SNE insufficient for clustering comparison.**
>
> We supplement with: (1) MMD/SWD metrics above showing strong alignment-gFID correlation; (2) Figure 4's systematic comparisons across 8 configurations --- DINO reference (4a) shows class-separated clusters, DMVAE (4e) replicates this, while $\beta$-VAE (4l) and data-independent priors (4j,k) show diffuse distributions.
>
> **W6: "SSL is good" based only on 10 classes.**
>
> This is a misunderstanding. **The DINO entry in Table 1 uses full ImageNet (1000 classes, ~1.28M images)**, achieving gFID 13.1. Table 3's main results also use full-ImageNet DINO (gFID 1.82/1.64). "SubDino" is an **intentionally degraded ablation** (10 classes, gFID 37.9) that *reinforces* our conclusion.
>
> **W7: Novelty --- similar to Pointwise Matching.**
>
> We respectfully disagree. **Pointwise Matching** constrains individual pairs $E(x_i) \to \phi(x_i)$ but places no constraint on collective density $q(z)$. **DMVAE** constrains $q(z)$ to match $p_r(z)$ at the density level with no per-sample correspondence. Per-sample losses can be "deceptively minimized" (Sec. 3); distributional constraints cannot. The VAE-baseline vs. Gaussian DMVAE comparison validates this --- same Gaussian target, same architecture:
>
> | Method | MMD | SWD | gFID |
> |---|---|---|---|
> | VAE-baseline (per-sample) | 0.09 | 1.06 | 27.3 |
> | Gaussian DMVAE (dist-level) | 0.07 | 0.81 | 26.6 |
>
> Additionally, DMVAE can target distributions with no per-sample correspondence (GMM, Gaussian), unlike Pointwise Matching which requires paired $\phi(x)$. Beyond this distinction, the paper contributes: (1) the first application of score-based distribution matching to aggregate posterior control, (2) the first systematic study of how different latent priors affect diffusion modeling, and (3) the theoretical analysis in Sec. 3 on why per-sample constraints are insufficient.
>
> **W8a:** We will clarify "complex and irregular" as multi-modal density with disconnected high-density regions and low-density "holes" (Rezende & Viola, 2018).
>
> **W8b:** Delta-function collapse ($\beta=10^{-6}$) is the **standard regime** of LDM, not a corner case. Under this regime, $q(z)$ becomes an empirical distribution on $\{E(x): x \sim p(x)\}$, far from Gaussian. Even with non-negligible $\beta$, $q(z)$ is a mixture of Gaussians with class-structured means --- multi-modal, not unimodal.
>
> **W8c/d:** We will define $\mathcal{L}_{\text{align}}$, "modeling efficiency," and standardize to "modeling" (American English).

---

### Official Review · Reviewer_2jf9 · 2026-03-16

**Soundness:** 3
**Presentation:** 4
**Significance:** 3
**Originality:** 4
**Overall Recommendation:** 4
**Confidence:** 4

**Summary:**

This paper studies the choice of distributions of the VAE latent space and their effects on downstreaming diffusion modelling. The authors propose a distribution matching approach to explicitly match the latent distribution to a given distribution on a distribution level, rather than on single sample level, particularly, by distribution matching distillation (DMD) techniques. The authors extensively benchmark different distributions to align for latent space representations and their differences on reconstruction and following diffusion modeling. ImageNet benchmark shows the diffusion models trained on DM-VAE can reach competitive results with less training epochs.

**Compliance With Llm Reviewing Policy:**

Affirmed.

**Key Questions For Authors:**

See weaknesses.

**Limitations:**

Yes.

**Strengths And Weaknesses:**

Strengths:

1. The presentation and writing of this papre is excellent and straightforward. This paper is well motivated with a clear goal, which distribution to align with VAE latents and which one is better for diffusion modeling. One can reproduce DM-VAE training following the authors detailed experiment settings.

1. This paper presents a new and interesting method, to use DMD to align VAE latents to a given distribution, and studies effects on following diffusion training given what distribution to align. The resulted model, can achieve competitive results with less training budgets, showing the convergence efficiency of DM-VAE latents. The authors also perform extensive ablation studies on how to align distributions and what to align.

Vulnerabilities:

1. The logic connection between "aligning VAE latents to a distribution" and "easy-to-diffusion" seems too weak. Why the distribution matching can VAE latents to be easy-to model? Also, it lacks empirical evidences on why distribution-level matching is necessary (or, performance ablation comparison to sample-wise alignment is missing).

1. According to the appendix, the authors use a 675M diffusion transformer to model the teacher score. I am wondering whether the model of this parameter count is competent for complicated representation distribution, like DINOv2, since after all it is a high-dimensional distribution? If the teacher score is not well learned or incur information loss, the motivation of the distribution matching is somewhat unjustified.

1. Could authors explain more detailed the performance difference between VAE-baseline (KL-regularized) and Gaussian entry? Seemed they are equivalent when apply distribution matching with Gaussian as a target. In the final result table, why are there CFG results missing (also visualizations in the appendix)? Is diffusion models trained on DM-VAE latents is not so compatible with classifier free guidance?

---

> ### Author Rebuttal · Authors · 2026-03-30
>
> # Response to Reviewer 2jf9
>
> We thank Reviewer 2jf9 for recognizing the novelty of using DMD to align VAE latents and the extensive ablation studies. We address the questions below:
>
> ---
>
> **Q1: Logic connecting distribution matching to easier diffusion modeling.**
>
> **Theoretical argument.** A diffusion model must learn $\nabla_z \log q(z_t)$, whose complexity depends on global geometric properties of $q(z)$. Per-sample losses (KL or MSE) constrain only individual codes but leave the aggregate geometry unregulated. As analyzed in Sec. 3, per-sample losses can be "deceptively minimized" while producing a topologically damaged aggregate posterior. Distribution-level matching directly sculpts $q(z)$ toward a reference $p_r(z)$, preventing such failure modes.
>
> **Empirical evidence.** Table 1 contains a direct comparison: VAE-baseline (per-sample KL to $\mathcal{N}(0,I)$) vs. Gaussian DMVAE (distribution-level matching to same target). Same target, same architecture. We report new distribution matching metrics:
>
> | Method | MMD $\downarrow$ | SWD $\downarrow$ | gFID-5k $\downarrow$ |
> |---|---|---|---|
> | VAE-baseline (per-sample KL) | 0.09 | 1.06 | 27.3 |
> | Gaussian DMVAE (dist-level) | 0.07 | 0.81 | 26.6 |
>
> Distribution-level matching achieves better alignment (lower MMD/SWD) and better gFID, validating that the *mechanism* of alignment matters, not just the choice of target.
>
> Furthermore, across all DMVAE ablation settings with the DINO prior, distributional alignment quality consistently predicts generation performance:
>
> | Setting | gFID-5k $\downarrow$ | MMD $\downarrow$ | SWD $\downarrow$ |
> |---|---|---|---|
> | $\lambda_{DM}$ = 1 | 16.7 | 0.243 | 1.212 |
> | $\lambda_{DM}$ = 10 (default) | 13.1 | 0.055 | 0.506 |
> | $\lambda_{DM}$ = 20 | 12.5 | 0.026 | 0.421 |
> | $\lambda_{DM}$ = 100 | 12.6 | 0.014 | 0.320 |
>
> Stronger distributional alignment consistently leads to better gFID. The slight gFID increase at $\lambda=100$ despite lower MMD/SWD reflects the reconstruction--matching trade-off (PSNR drops from 21.4 to 19.6).
>
> ---
>
> **Q2: Teacher score model capacity (675M) for DINOv2 distribution.**
>
> We believe this concern stems from a dimensional misunderstanding. The teacher operates on the **projected $d=32$ latent space**, not the original DINOv2 feature dimension. As described in Appendix B.1--B.2, all representations are projected to $d=32$ per spatial token, so the teacher models a $16 \times 16 \times 32 = 8{,}192$-dimensional distribution. A 675M model is substantially overparameterized for our task.
>
> The score network size ablation (Table 2) confirms: Small (100M) achieves gFID 13.9, L (400M) 13.1, XL (675M) 12.5. The diminishing returns confirm even a small network captures the $d=32$ distribution adequately. DMVAE's end-to-end gFID of 1.64, surpassing VA-VAE (2.17) and AlignTok (2.04) which use the same DINOv2 features via pointwise alignment, validates the teacher preserves sufficient distributional information. We will add dimensionality clarifications in the revision.
>
> ---
>
> **Q3a: VAE-baseline vs. Gaussian DMVAE.**
>
> These are *not* equivalent, and their gap validates our central thesis. **VAE-baseline** uses per-sample KL with $\beta \approx 10^{-6}$ (Rombach et al., 2022), which is nearly vacuous --- posteriors collapse to delta functions and the aggregate $q(z)$ becomes a complex empirical distribution far from Gaussian. **Gaussian DMVAE** applies distribution-level matching via Eq. (8), directly penalizing global geometric deviations. As shown in Q1, Gaussian DMVAE achieves lower MMD (0.07 vs. 0.09) and SWD (0.81 vs. 1.06), confirming distribution-level matching produces a genuinely more Gaussian aggregate posterior. The level of constraint matters even for the same target.
>
> ---
>
> **Q3b: Missing CFG results.**
>
> Table 3 reports without-CFG to isolate latent space quality. The CFG weight in Table 2 (3.0/5.0) refers to guidance during DM loss computation (Eq. 8), not final generation. DMVAE is **fully compatible with CFG**:
>
> | Epochs | w/o CFG | w/ CFG |
> |---|---|---|
> | 64 | 3.22 | 2.40 |
> | 400 | 1.82 | 1.54 |
> | 800 | 1.64 | 1.49 |
>
> **We do not compare CFG results with baselines because they use method-specific tricks**: VA-VAE applies CFG only to the first 3 channels, RAE uses autoguidance requiring additional model training and guidance interval tuning. **All of the above introduces significant tuning complexity beyond standard CFG, making direct comparison misleading.**

---

> > ### Author Rebuttal · Reviewer_2jf9 · 2026-04-04
> >
> > Thanks for the authors' rebuttal. I will maintain my positive score.

---

### Decision · Program_Chairs · 2026-04-30

**Decision:**

Accept (regular)

**Comment:**

The authors successfully resolved most concerns raised by the reviewers, reaching to unanimous acceptance recommendations by them. One reviewer (G7gD) expressed a remaining concern regarding the fairness of hyperparameter tuning across priors and the mechanistic explanation, but this AC agrees that this can be further addressed in the camera-ready rather than going through another full review process. We strongly encourage the authors to add comprehensive evidence regarding this in the final version.